# Spatial and longitudinal tracking of enhancer-AAV vectors that target transgene expression to injured mouse myocardium

David W Wolfson[1,2], Joshua A Hull[2], Yongwu Li[3], Trevor J Gonzalez[2], Mourya D Jayaram[2], Garth W Devlin[2], Valentina Cigliola[1,4], Kelsey A Oonk[1], Alan Rosales[3], Nenad Bursac[3], Aravind Asokan[2]*, Kenneth D Poss[1,5,6]*

[1]Department of Cell Biology, Duke Regeneration Center, Duke University School of Medicine, Durham, United States; [2]Department of Surgery, Duke University School of Medicine, Durham, United States; [3]Department of Biomedical Engineering, Duke University, Durham, United States; [4]Department of Pharmacology, Vanderbilt University, Nashville, United States; [5]Morgridge Institute for Research, Madison, United States; [6]Department of Cell and Regenerative Biology, University of Wisconsin-Madison, Madison, United States

*For correspondence:
aravind.asokan@duke.edu (AA);
kposs@morgridge.org (KDP)

**Competing interest:** The authors declare that no competing interests exist.

## eLife Assessment

This study identifies novel approaches to improving transgene expression in the injured mammalian myocardium through a combination of a tissue regeneration enhancer element and engineered AAVs - specifically, a liver-detargeting capsid, AAV.cc84, and an in vivo library screen-selected AAV-IR41. The evidence is **convincing**, and the AAV vectors are of **fundamental** value to the field of cardiac gene therapy. Future research exploring how to combine the features of AAV.cc84 and AAV-IR41 could yield an even more promising vector for therapeutic use.

**Abstract** Tissue regeneration enhancer elements (TREEs) direct expression of target genes in injured and regenerating tissues. Additionally, TREEs of zebrafish origin were shown to direct expression of transgenes in border zone regions after cardiac injury when packaged into recombinant adeno-associated viral (AAV) vectors and introduced into mice. Future implementation of TREEs into AAV-based vectors as research tools and potential gene therapy modalities requires a deeper understanding of expression dynamics and potential off-target effects. Here, we applied in vivo bioluminescent imaging to mice systemically injected with AAV vectors containing different combinations of capsids, enhancers, and timing of delivery. Longitudinal tracking of expression directed by different TREEs revealed distinct amplitudes and durations of reporter gene expression in the injured heart. The liver-de-targeted AAV capsid, AAV.cc84, could deliver TREEs either pre- or post-cardiac injury to negate off-target expression in the liver while maintaining transduction in the heart. By screening AAV9-based capsid libraries dosed systemically in mice post-cardiac injury, we discovered a new capsid variant, AAV.IR41, with enhanced transduction in cardiac injuries and with elevated transduction of TREE-driven transgenes versus conventional AAV9 vectors. In vivo bioluminescence imaging offers insights into how enhancers and engineered capsids can be implemented to modulate spatiotemporal transgene expression for targeted therapies.

## Introduction

The inability of the adult mammalian heart to regenerate after an infarction injury has motivated attempts to discover factors with cardiomyogenic, angiogenic, or anti-fibrotic potential. Many genes have been implicated by genetic studies in zebrafish and mouse to elevate or restrict cardiomyocyte (CM) proliferation after cardiac injury, with groups targeting factors controlling differentiation, chromatin state, metabolism, and other events (*Eulalio et al., 2012*; *Choi et al., 2013*; *Fang et al., 2013*; *Mahmoud et al., 2013*; *Karra et al., 2015*; *Bassat et al., 2017*; *Toischer et al., 2017*; *Hirose et al., 2019*; *Chen et al., 2021*; *Li et al., 2023*; *Nguyen et al., 2023*). Notable mitogenic influences include the secreted factor Neuregulin1 (Nrg1), signaling through ErbB family receptors (*Bersell et al., 2009*; *D'Uva et al., 2015*; *Gemberling et al., 2015*), the transcription factor (TF) Klf1 (*Ogawa et al., 2021*), and Yap/Taz factors, normally restrained from the nucleus by Hippo in CMs (*von Gise et al., 2012*; *Heallen et al., 2013*; *Xin et al., 2013*; *Morikawa et al., 2015*; *Bassat et al., 2017*; *Leach et al., 2017*; *Morikawa et al., 2017*; *Monroe et al., 2019*). The signaling environment is complex, with epicardium, endocardium, nerves, macrophages, T cells, vasculature, and oxygen levels each influencing CM regeneration (*Lepilina et al., 2006*; *Kim et al., 2010*; *Kikuchi et al., 2011*; *Aurora et al., 2014*; *Mahmoud et al., 2015*; *Wang et al., 2015*; *Hui et al., 2017*; *Nakada et al., 2017*; *Sugimoto et al., 2017*; *Zhao et al., 2019*; *Xia et al., 2022*; *Peterson et al., 2024*; *Shin et al., 2024*; *Sun et al., 2024*; *Weinberger et al., 2024*).

It has become crucial to decipher how genes involved in regenerative events engage upon injury in CMs and other cardiac cells. Recently, distal regulatory elements referred to as tissue regeneration enhancer elements (TREEs) were described that direct gene expression in regenerating zebrafish hearts, a new concept for the field of regenerative biology (*Kang et al., 2016*). TREEs have been identified by chromatin profiling in CMs, epicardium, and other tissues of zebrafish like spinal cord and fins (*Goldman et al., 2017*; *Thompson et al., 2020*; *Cao et al., 2022*; *Cigliola et al., 2023*), as well as in a variety of other species, tissues, and injury contexts (*Harris et al., 2016*; *Vizcaya-Molina et al., 2018*; *Gehrke et al., 2019*; *Harris et al., 2020*; *Wang et al., 2020*; *Jimenez et al., 2022*; *Sun et al., 2022*; *Suzuki et al., 2022*; *Li et al., 2023*; *Sun and Poss, 2023*; *Zlatanova et al., 2023*; *Kawaguchi et al., 2024*). Critically, TREEs of zebrafish origin can be recognized by the transcriptional machinery of small (mice) or even large (pigs) mammals. Systemic introduction of adeno-associated viral (AAV) vectors with TREE-cargo cassettes can spatiotemporally target factor delivery to the injured mammalian heart or spinal cord (*Cigliola et al., 2023*; *Yan et al., 2023*). For instance, a TREE paired with a heavily mutated YAP transgene transiently directs expression to the border zone of a mouse heart injured by ischemia/reperfusion (I/R), stimulating localized CM cycling and functional recovery (*Yan et al., 2023*). Thus, short TREE sequences found in zebrafish, and presumably elements with similar properties from other organisms, can act as targeting vehicles for regenerative augmentation.

Critical remaining questions include how TREEs interact with target genes and TFs, and how they can be harnessed for safe regenerative applications. Translational use of TREEs for delivering pro-regenerative factors requires optimized delivery systems to maximize efficacy and safety while minimizing off-target risks. Recently, new engineered and variant forms of AAV capsids have been developed to modulate tissue tropism and cell type specificity, such as liver de-targeting with AAV. cc84 (*Gonzalez et al., 2023*). Here, we performed a study to assess how choice of TREE and AAV capsid can influence the spatiotemporal control of transgene expression in mouse models of myocardial infarction (MI). Using bioluminescence imaging, we mapped and longitudinally tracked AAV transgene expression in a whole-body manner and screened for new variant AAV9 capsids with enhanced tropism for infarcted myocardial tissue over healthy myocardium.

## Results and discussion

### IVIS detects TREE-directed gene expression after myocardial injury

We previously showed that TREEs are capable of directing AAV transgene expression in an injury-responsive manner in mouse models of spinal cord and cardiac damage (*Cigliola et al., 2023*; *Yan et al., 2023*). In these studies, TREEs were placed upstream of a murine *Heat shock protein 1a* (*Hspa1a*) promoter, normally silent in mouse tissues, and an EGFP reporter transgene, and AAV vectors were delivered systemically. EGFP transgene expression to observe TREE activity could not be

longitudinally tracked in individual mice, limiting our ability to observe whole-body spatial distribution of expression (on- and off-target).

To longitudinally monitor AAV transgene expression using in vivo bioluminescence imaging (IVIS), albino BALB/c mice were injected intravenously with AAV9 containing either a chicken beta actin (CBA) or *Hspa1a* promoter upstream of a firefly luciferase (*fLuc*) transgene (*Figure 1—figure supplement 1A*). As expected, the strong, constitutively active CBA promoter directed robust fLuc expression across the mouse body, whereas mice transduced with AAV9 harboring *Hspa1a::fLuc* had relatively minimal luminescence, mostly originating from the liver (*Figure 1—figure supplement 1B*). Luminescence in the heart, liver, and whole-body plateaued and remained stable from 30 to 68 days post-AAV delivery for both CBA and *Hspa1a* vectors, as indicated by average radiance measurement (*Figure 1—figure supplement 1C–E*).

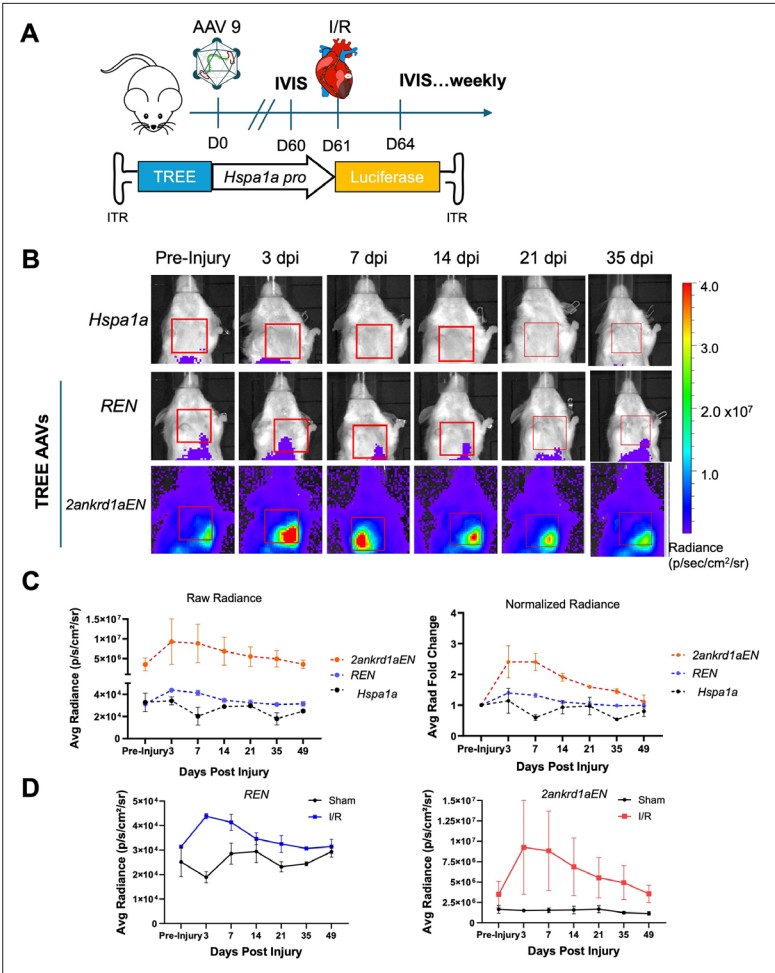

**Figure 1.** Longitudinal tracking of tissue regeneration enhancer element (TREE)-directed transgene expression in injured mice by in vivo bioluminescence imaging (IVIS). (**A**) Schematic illustration of study design. Albino BALB/c mice were systemically injected with AAV9 vectors packaging fLuc reporter cassettes directed by TREEs and a permissive promoter. Mice underwent ischemia/reperfusion (I/R) surgery at D61 and were imaged by IVIS at indicated time points. (**B**) Representative IVIS images indicate changes of expression over time and space for each vector. Cardiac region of interest (ROI) indicated by red box. n=2 mice. (**C**) Average radiance measured from cardiac ROIs plotted over days post-injury (dpi). Average radiance normalized to their baseline pre-injury was also plotted (right). n=2 mice. (**D**) Average cardiac radiance showed a transient increase in expression for both *REN* (left) and *2ankrd1aEN* (right) after I/R injury, whereas sham-operated animals showed relatively constant expression. n=2 mice for I/R, n=3 mice for sham.

The online version of this article includes the following figure supplement(s) for figure 1:

**Figure supplement 1.** In vivo bioluminescence imaging (IVIS) imaging for tracking spatiotemporal expression of rAAV vectors.

To determine if IVIS detected gene expression directed by zebrafish TREE sequences in injured mouse tissues, we then systemically delivered AAV9 vectors containing *Hspa1a*::fLuc with or without a TREE sequence 60 days prior to I/R or sham surgery. This 60-day period was intended to allow AAV expression to stabilize prior to surgery (*Figure 1A*). We employed two TREEs that previously indicated different injury responsiveness and temporal dynamics by histology, the *runx1*-linked enhancer (*REN*) and the *2ankrd1a*-linked enhancer *2ankrd1aEN* (*Goldman et al., 2017*). Transduction of either TREE sequence resulted in a marked increase in bioluminescence signal in the thoracic cage overlying the heart. Bioluminescence peaked within the first 3 days after I/R and returned to pre-injury baseline levels within 35 days post-injury (dpi) (*Figure 1B and C*; n=2 mice). Cardiac bioluminescence showed no change after sham surgery for all mice, indicating that TREE-directed expression occurred in response to myocardial trauma (*Figure 1D*). Notably, each AAV vector (CBA, *Hspa1a*, and TREE) demonstrated off-target expression in the abdominal region directly above the liver after transduction, which was

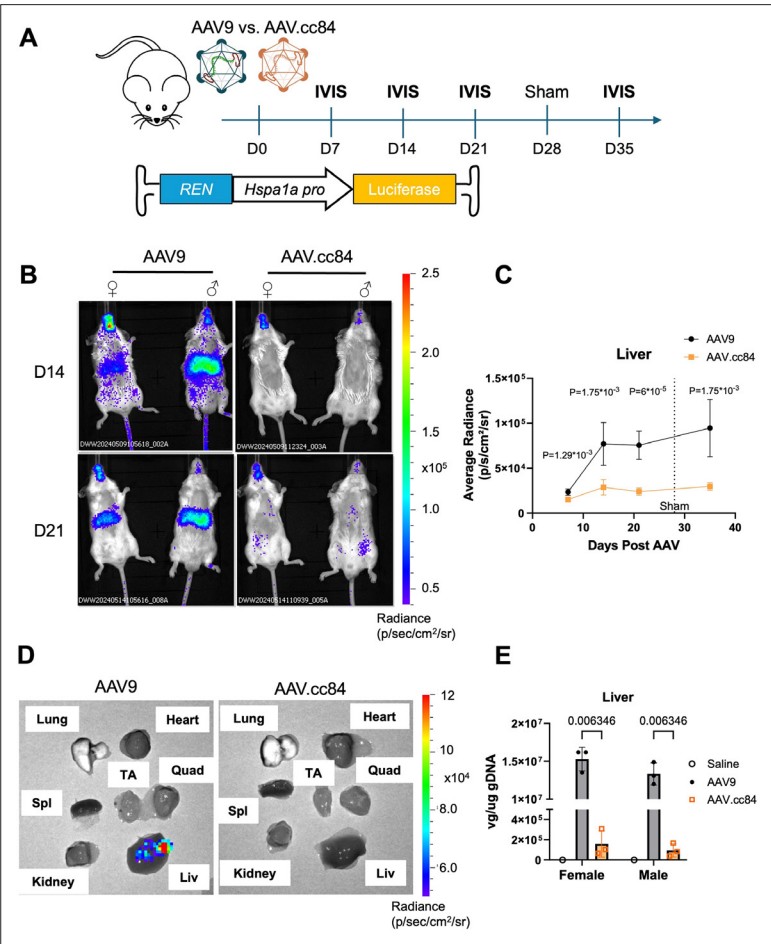

**Figure 2.** Liver-de-targeted AAV.cc84 capsid limits hepatic expression from tissue regeneration enhancer elements (TREEs). (**A**) Schematic of experimental timeline, comparing AAV9 and AAV.cc84 capsids for systemic delivery of *REN-Hsp1a*::fLuc. Mice were in vivo bioluminescence imaging (IVIS) imaged in the weeks following AAV delivery and post-sham surgery. (**B**) Representative IVIS images of mice injected with either AAV9 (left) or AAV.cc84 (right) at 14 (top) and 21 days (bottom) post-AAV injection. Mice were also subdivided by biological sex to account for sex differences in AAV liver tropism. (**C**) Average radiance from liver regions of interest (ROIs) showed significantly higher expression in AAV9-transduced mice compared to AAV.cc84 through all timepoints (n=6 mice, Holm–Sidak multiple comparisons test). (**D**) Representative IVIS images of harvested organs at 42 days post-AAV demonstrate liver expression with AAV9 (left) while undetected with AAV.cc84 (right). (**E**) Vector genome quantification from collected liver samples reveals higher liver transduction with AAV9 compared to AAV.cc84 for both female and male mice.

The online version of this article includes the following figure supplement(s) for figure 2:

**Figure supplement 1.** AAV.cc84 capsid retains cardiac tropism while minimizing liver transduction.

also indicated to a limited degree by histology in our previous study (*Figure 1B*, *Figure 1—figure supplement 1B*; *Yan et al., 2023*).

## Live imaging of a liver de-targeted AAV9 capsid variant

To attempt to reduce hepatic transgene expression while maintaining cardiac tropism, we packaged *REN-Hspa1a::fLuc* into AAV.cc84, a recently discovered AAV9 variant with reduced liver transduction (*Gonzalez et al., 2023*), as well as AAV9 for comparison. *REN* was selected as the liver luminescence was more prominent with respect to the cardiac signal in the above experiments. Following AAV delivery, mice were imaged weekly and also given a sham surgery at day 28, to assess spatiotemporal dynamics of bioluminescence (*Figure 2A*). IVIS images revealed off-target luminescence in male and female livers with AAV9, whereas AAV.cc84 showed no detectable signal in the liver for all sexes of mice over 35 days following AAV (*Figure 2B*, *Figure 2—figure supplement 1A*). Quantification indicated that AAV9 enabled substantially higher expression in liver than AAV.cc84, which remained minimal over time (*Figure 2C*, n=6 mice). Luminescence in the heart was similar between AAV9 and AAV.cc84, and, as expected, luminescence showed no change upon sham surgery (*Figure 2—figure supplement 1B*). Six weeks post-AAV, while AAV.cc84-injected mice continued to show minimal expression across many biopsied organs, AAV9-injected mice showed off-target expression in the liver (*Figure 2D*). Vector genome quantification by qPCR revealed AAV9-injected mice contained a much higher viral load in liver tissue compared to AAV.cc84, for both males and females (*Figure 2E*, n=3 mice). Vector genomes in the heart were similar between AAV9 and AAV.cc84 for both sexes of mice (*Figure 2—figure supplement 1C*). In addition to heart and liver, luminescence was also observed in the head/neck and lower abdomen regions, which showed no difference between AAV9 and AAV.cc84 (*Figure 2—figure supplement 1D and E*). These results indicate that AAV.cc84 can be implemented to minimize off-target liver expression of TREEs, evading liver transduction while maintaining cardiac tropism.

## Live imaging of post injury, liver de-targeted TREE-AAV transduction

Therapeutic TREE use would require their delivery after injury occurs. To examine expression dynamics in a therapeutically relevant context, we systemically delivered the liver-detargeted vector AAV.cc84 carrying *2ankrd1aEN-Hspa1a::fLuc* or *Hspa1a::fLuc* at 4 dpi in mice given an I/R injury (*Figure 3A*). Echocardiograms to measure cardiac ejection fraction before and after I/R were used to estimate and compare injury size in both treatment groups (*Figure 3—figure supplement 1A*). *2ankrd1aEN-Hspa1a::fLuc*-transduced mice showed robust fLuc activity in the thoracic cage overlying the heart, which increased up to 28 dpi and plateaued afterwards (*Figure 3B and C*; n=4 mice). Conversely, cardiac bioluminescence in *Hspa1a::fLuc*-transduced mice remained relatively minimal over time (*Figure 3B and C*, *Figure 3—figure supplement 1B*; n=4 mice). IVIS imaging revealed a baseline of *2ankrd1aEN*-directed cardiac expression in sham-operated mice (*Figure 3—figure supplement 1C*), which was elevated approximately threefold in mice with I/R injury compared to sham at 7 dpi (*Figure 3D*, n=4 mice). *2ankrd1aEN*-directed bioluminescence in the heart increased during the first 28 days for sham-operated mice, likely indicative of increased AAV transduction and transgene expression over time (noting that AAV deliveries in *Figure 1* experiments were 60 days prior to luminescence measurements; *Figure 3E*). Notably, I/R injured mice maintained higher cardiac activity throughout the 63-day time course, with the greatest differences at 7 and 21 dpi (*Figure 3E*). Together, these results indicate that *2ankrd1aEN*-directed gene expression can be estimated longitudinally over long time periods, and in agreement with our previous histological analysis, *2ankrd1aEN* directs long-lasting gene expression in the injured mouse heart.

## In vivo library screen for AAV9 capsid variants with increased targeting of injury sites

Directed evolution of AAV libraries has been previously used to discover new capsids with improved functional properties, such as liver de-targeting (AAV.cc84), enhanced potency (AAV.cc47), and tropism for specific cell types (AAV.ark313, AAV.k20) (*Gonzalez et al., 2022*; *Gonzalez et al., 2023*; *Nyberg et al., 2023*; *Nyberg et al., 2025*; *Rosales et al., 2025*). We reasoned that this approach may be useful to identify variants with enhanced tropism for infarcted myocardium, which has altered extracellular matrix and cell type landscapes.

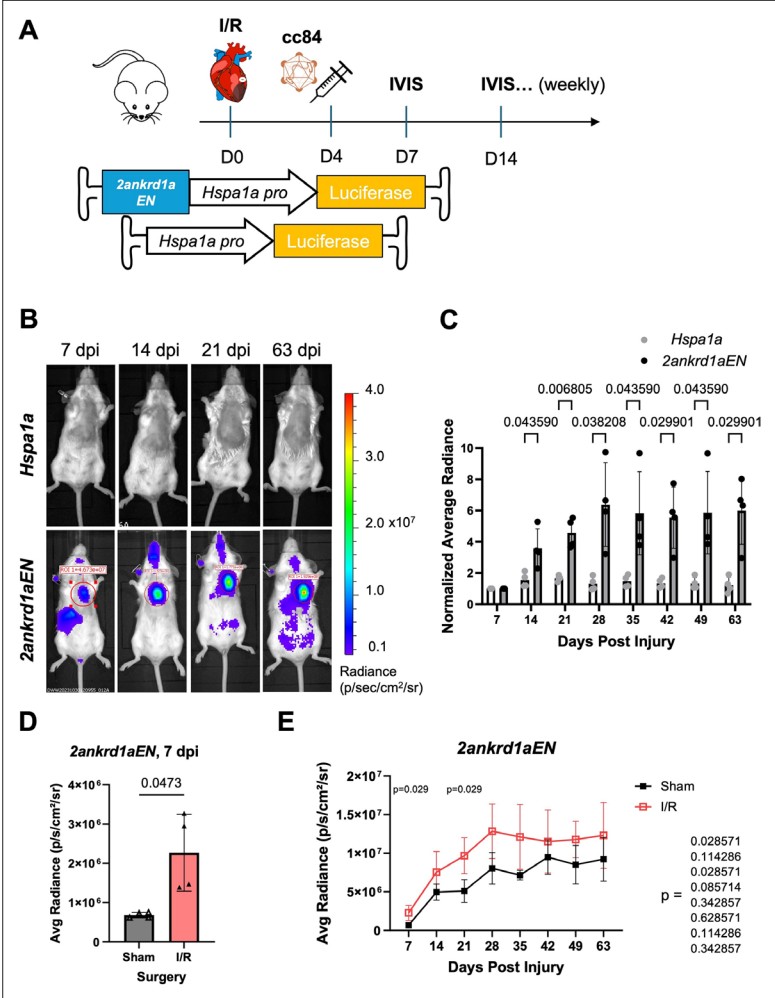

**Figure 3.** Post-injury delivery of AAV.cc84-packaged *2ankrd1aEN* targets expression to cardiac injuries. (**A**) Schematic of experimental timeline comparing expression between *Hsp1a* and *2ankrd1aEN* when delivered at 4 dpi. (**B**) Representative in vivo bioluminescence imaging (IVIS) images of mice injected with *Hsp1a* (top) or *2ankrd1aEN- Hsp1a* (bottom) after ischemia/reperfusion (I/R) injury. (**C**) Cardiac average radiance normalized to the 7 dpi time point increased over time with *2ankrd1aEN* while remaining stable with *Hsp1a* (n=4 mice, Holm–Sidak multiple comparisons test). (**D**) Average cardiac radiance directed by *2ankrd1aEN* was significantly higher in mice with I/R injury compared to sham at 7 dpi (n=4 mice, Welch's *t*-test). (**E**) Average cardiac radiance was more significantly elevated in the first 21 days post-injury in mice with I/R injury compared to sham (n=4 mice, Mann–Whitney tests).

The online version of this article includes the following figure supplement(s) for figure 3:

**Figure supplement 1.** Delivery of AAV.cc84 packaged with *2ankrd1aEN* after myocardial injury.

To find capsid variants with enhanced tropism for infarcted myocardial tissue over healthy myocardium, we incorporated previously described AAV capsid libraries with saturation mutagenesis at the variable region IV epitope (***Gonzalez et al., 2023***). A previous report indicated that AAV9 is observed to preferentially transduce border zones of infarcted myocardium when delivered acutely (10 min to 3 dpi) after injury, which was speculated to be driven by a combination of increased capillary permeability, sialidase activity, and DNA damage after cardiac ischemia (***Konkalmatt et al., 2012***). We attempted to limit the effects of these acute changes by delivering capsid libraries to mice that had undergone sham or I/R surgery at 9 dpi, when the acute inflammation phase is complete and the repair phase with myofibroblast activation and proliferation has begun (***Figure 4A***, n=2 mice). At 11 dpi, cardiac tissue was harvested and separate biopsies were acquired from infarcted or remote myocardium. Viral genomes from each biopsy were amplified and sequenced, and infarcted tissue

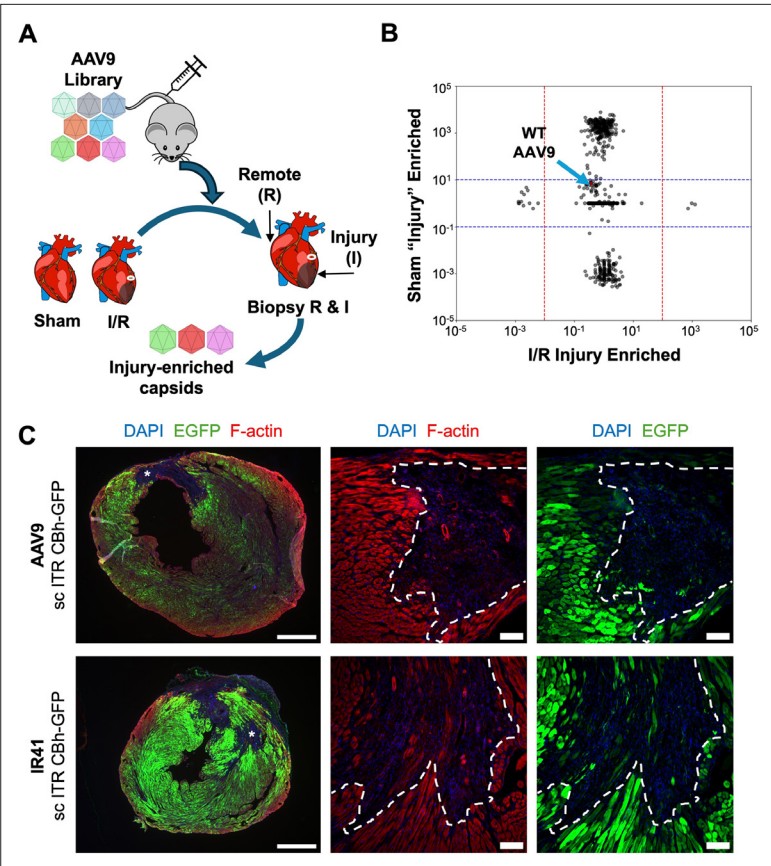

**Figure 4.** Screening of AAV libraries for enriched capsids in injured myocardium. (**A**) Schematic of AAV capsid library screening delivered systemically to mice with either sham (n=1 mouse) or ischemia/reperfusion (I/R) (n=2 mice) injury at 9 dpi. Two days later, hearts were biopsied to collect AAV genomes in either injured or remote tissues. (**B**) Capsid sequenced reads enriched in the injured tissues were plotted against sham animals. Each point represents a unique capsid. Wild-type AAV9 capsid is marked by blue arrow. (**C**) Representative fluorescence imaging of AAV9 (top) or variant capsid IR41 (bottom) delivering a self-complementary CBA::EGFP cassette at 16 dpi. Asterisks indicate infarct site, imaged at higher magnification in middle and right panel. Dashed white lines indicate the border zone region. Left scale bar, 1 mm. Middle and right scale bar, 100 um.

The online version of this article includes the following figure supplement(s) for figure 4:

**Figure supplement 1.** Screening AAV capsid libraries enriched in injured myocardium.

---

fold-change over remote tissue was calculated from mapped reads for each capsid variant (*Figure 4B*). We found three unique capsids that were enriched in the infarcted tissue over remote for both I/R mice, while being comparably expressed in both biopsies of the sham mouse (*Figure 4B*, *Figure 4— figure supplement 1A*). Of the three variant capsids, we found the capsid with the highest number of reads in infarcted tissue (AAV.IR41) transduced I/R-injured hearts more effectively than AAV9 and the other two variant capsids (IR42, IR43) when packaged with a CBA::EGFP transgene (*Figure 4C*, *Figure 4—figure supplement 1B*). When systemically delivered at 9 dpi, IR41 transduced larger areas of myocardium surrounding areas of infarct compared to AAV9 (*Figure 4C*). Using F-actin staining to mark the border zone, AAV9 and IR41 primarily transduced CMs directly at the border zone with similar effectiveness (*Figure 4C*). EGFP did not notably co-localize with vimentin staining of non-myocytes for either AAV9 or IR41, thus both capsids still primarily transduce CM (*Figure 4—figure supplement 1C*). Thus, sequence-based screening identified a capsid with enhanced transduction and expression in injured myocardium.

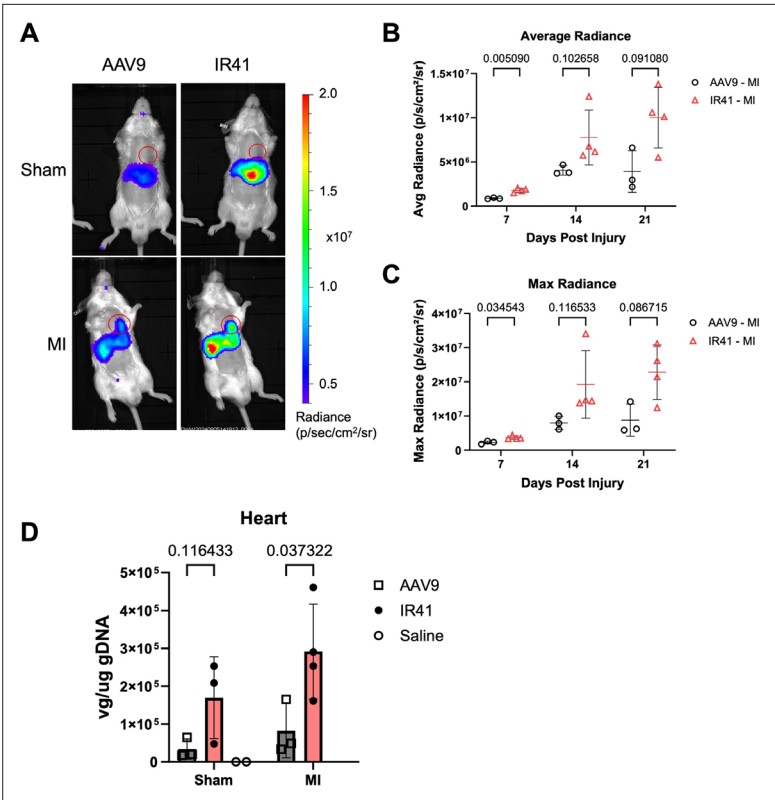

**Figure 5.** Post-injury delivery of AAV.IR41 variant capsid enhances *2ankrd1aEN*-directed expression in injured myocardium. (**A**) Representative in vivo bioluminescence imaging (IVIS) images of mice with sham (top) or (bottom) surgery transduced with either AAV9 (left) or IR41 (right) packaged with *2ankrd1aEN- Hsp1a::*fLuc (n=3–4 mice). (**B, C**) Cardiac average (**B**) and maximum (**C**) radiance was elevated in MI mice transduced with IR41 compared to AAV9 (n=3–4 mice, Holm–Sidak multiple comparisons test). (**D**) Viral genome quantification from heart tissues was elevated in MI mice injected with IR41 compared to AAV9 (n=3–4 mice, Holm–Sidak multiple comparisons test).

The online version of this article includes the following figure supplement(s) for figure 5:

**Figure supplement 1.** Post-injury delivery of variantAAV.IR41 variant capsid enhances *2ankrd1aEN*-directed expression in injured myocardium over AAV9.

## IR41 capsid delivery of TREE-directed transgene expression in injured hearts

To assess the efficacy of TREE-directed transgene targeting using IR41 capsid, we systemically transduced mice at 4 days after permanent left coronary artery ligation (MI) with either AAV9 or IR41 harboring a *2ankrd1aEN-Hspa1a::*fLuc transgene. IVIS imaging revealed higher expression levels in animals transduced with IR41 compared to AAV9, in both sham and MI groups (*Figure 5A*, *Figure 5— figure supplement 1A*; n=3 mice). We also noted higher hepatic expression in both sham and infarcted mice with IR41 compared to AAV9 (*Figure 5A*), although no new tissue transduction was introduced as detectable by biodistribution and reporter gene expression. The average and maximum radiances recorded in IR41-transduced mouse hearts were significantly higher than AAV9 by 7 dpi and remained elevated at 21 dpi (*Figure 5B and C*). In mice that underwent sham surgery, IR41 had a higher average and maximum radiance than AAV9 at 7 dpi alone but was relatively similar at 14 and 21 dpi (*Figure 5— figure supplement 1B and C*). Vector genome quantification by qPCR showed that IR41 had roughly twice as high viral uptake than AAV9 in mice given cardiac injuries (*Figure 5D*, n=3 mice). Interestingly, IR41-transduced mice showed a trend toward higher cardiac viral genome loads than AAV9-transduced mice upon sham injury, indicating that IR41 might increase tropism in uninjured hearts, but its effects will be greatest when delivered to injured hearts (*Figure 5D*). Together, our experiments indicate that increased tropism of AAV.IR41 for injured myocardium, defined by enhanced uptake and

higher transgene expression versus control AAV9, can be complemented by injury-responsive TREEs for targeting cargo delivery to sites of cardiac damage.

## Conclusions and limitations

Here, we report techniques for spatial and longitudinal tracking of injury- and regeneration-responsive, TREE-directed transgene expression from AAV vectors systemically infused into mice, either pre- or post injury. We characterized whole-body temporal dynamics of TREE-directed AAV expression and how expression may be modulated with the incorporation of engineered capsids, including those that may better target injured tissue. Our study indicates that IVIS imaging for assessing amplitude and specificity of TREE-directed transgene expression will aid future development of enhancer-based gene therapies relevant to cardiac regeneration and disease. With an IVIS imaging modality, effects of capsid alterations and enhancer sequence alterations can be efficiently and noninvasively assessed to refine their ability to target gene expression in the injured heart.

There are several limitations we point out from these techniques. Expression by internal organs must also be approximated with potentially subjective ROIs that cannot be verified until harvesting the organs at sacrifice. Although the expression is approximated, we were able to observe unexpected off-target TREE expression in the liver, neck, and abdomen, which was modulated by capsid engineering. Additionally, future work exploring cell surface markers or receptors is needed to better understand the apparent enhanced tropism of IR41 over AAV9, and to engineer liver detargeting attributes into this form. For instance, it is possible that combining the modifications in AAV.cc84 and AAV.IR41 would produce additive effects. Finally, larger animal models are not suited for IVIS imaging due to limitations of tissue penetrance, and findings in mice may not fully recapitulate those in larger animal models and humans. Our AAV capsid library screening was limited to mice, and capsid hits remain to be tested in other species. While further optimization is required, this work lays a foundation for refining TREE-targeted factor delivery to injured myocardium and expanding their use in applications for therapeutic tissue repair.

# Materials and methods

**Key resources table**

| Reagent type (species) or resource | Designation | Source or reference | Identifiers | Additional information |
|---|---|---|---|---|
| Strain, strain background (*Mus musculus*) | Male and female C57BL/6J | The Jackson Laboratory | Strain 000664; RRID:IMSR_JAX:000664 | |
| Strain, strain background (*M. musculus*) | Male and female BALB/c | Charles River | Strain 028; RRID:IMSR_CRL:028 | |
| Recombinant DNA reagent | AAV9: CBA::fLuc | This paper | | Available from corresponding authors |
| Recombinant DNA reagent | AAV9: *Hspa1a*::fLuc | This paper | | Available from corresponding authors |
| Recombinant DNA reagent | AAV9: REN-*Hspa1a*::fLuc | This paper | | Available from corresponding authors |
| Recombinant DNA reagent | AAV9: *2ankrd1aEN*-*Hspa1a*::fLuc | This paper | | Available from corresponding authors |
| Recombinant DNA reagent | AAV.cc84: REN-*Hspa1a*::fLuc | This paper | | Available from corresponding authors |
| Recombinant DNA reagent | AAV.cc84: *Hspa1a*::fLuc | This paper | | Available from corresponding authors |
| Recombinant DNA reagent | AAV.cc84: *2ankrd1aEN* -*Hspa1a*::fLuc | This paper | | Available from corresponding authors |
| Recombinant DNA reagent | AAV.IR41: *2ankrd1aEN* -*Hspa1a*::fLuc | This paper | IR41 VR4 peptide sequence: GPGVGAR | Available from corresponding authors |

*Continued on next page*

*Continued*

| Reagent type (species) or resource | Designation | Source or reference | Identifiers | Additional information |
|---|---|---|---|---|
| Recombinant DNA reagent | AAV.IR42: *2ankrd1aEN - Hspa1a*::fLuc | This paper | IR42 VR4 peptide sequence: ASRNVVT | Available from corresponding authors |
| Recombinant DNA reagent | AAV.IR43: *2ankrd1aEN - Hspa1a*::fLuc | This paper | IR43 VR4 peptide sequence: SDSQYVQ | Available from corresponding authors |
| Recombinant DNA reagent | AAV9 capsid library | *Gonzalez et al., 2022*; *Gonzalez et al., 2023* | | |
| Cell line (human) | HEK293 | ATCC; UNC Viral Vector Core; *Gonzalez et al., 2022*; *Gonzalez et al., 2023* | RRID:CVCL_0045 | Obtained from UNC Viral Vector Core for production of recombinant AAV at Duke, see Asokan lab citations |
| Antibody | Rabbit anti- GFP | Invitrogen | #A-11122; RRID:AB_221569 | 1:1000 |
| Antibody | Mouse anti-vimentin | DSHB | #40E-C; RRID:AB_528504 | 1:100 |
| Antibody | Goat anti-rabbit IgG Alexa Fluor 488 | Invitrogen | #A-11008; RRID:AB_143165 | 1:500 |
| Antibody | Goat anti- mouse IgG Alexa Fluor 555 | Invitrogen | #A21422; RRID:AB_2535844 | 1:500 |
| Sequence-based reagent | ITR F | This paper | PCR primers | 5' AACATG CTACGC AGAGAG GGAGTGG 3' |
| Sequence-based reagent | ITR R | This paper | PCR primers | 5'CATGAGA CAAGGA ACCCCT AGTGAT GGAG 3' |
| Sequence-based reagent | AAV9 lib amp F | This paper | PCR primers | 5' AGCACG GTCCAGGT CTTCAC 3' |
| Sequence-based reagent | AAV9 lib amp R | This paper | PCR primers | 5' ATGTCAG TCTAGAC CAAAGTT CAACTGA AACGAAT TAAACGG 3' |
| Sequence-based reagent | WPRE-bGH F | This paper | PCR primers | 5' CTTCGCC CTCAGAC GAGTCGGA 3' |
| Sequence-based reagent | WPRE-bGH R | This paper | PCR primers | 5' TGGCTGG CAACTAG AAGGCACA 3' |
| Commercial assay or kit | PureLink Genomic DNA mini kit | Invitrogen | K182002 | |
| Chemical compound, drug | D-luciferin, potassium salt | Gold Biotechnology | LUCK-1G | |

## Animals

All mouse protocols were approved by the Institutional Animal Care and Use Committee (IACUC) at Duke University (Protocols #A003-22-01, #A025-24-01), which is accredited by the Association for Assessment and Accreditation of Laboratory Animal Care-International (AAALAC). All mice were housed and maintained in the Duke University DLAR mouse facility. Wild-type male and female C57BL/6J (The Jackson Laboratory, Strain #000664) and BALB/c (Charles River, Strain #028) mice were used for this study.

## Ischemia/reperfusion and LAD ligation injury models

The procedure for I/R myocardial injury via temporary LCA ligation was adapted from previously described protocols (*Kaiser et al., 2005*). Briefly, adult mice, 8–12 weeks of age, were anesthetized with ketamine/xylazine, intubated, and placed on a rodent ventilator. The chest cavity was entered in the third intercostal space at the left lateral border. The left atrium was gently deflected out of the field to expose the left anterior descending artery. Coronary ligation was performed by tying an 8–0 nylon suture ligature around the proximal LAD artery. Occlusion of the LAD was confirmed by the anterior wall of the left ventricle turning a pale color. LAD occlusion was maintained for 40 min. After the ischemia period, the suture was untied, and reperfusion of the LAD was confirmed by the return of a pink-red color to the anterior wall. The procedures for LAD ligation were performed with the same approach but with permanent ligation of the left coronary artery by tying an 8–0 nylon suture ligature around the proximal LAD artery. Occlusion of the LAD was confirmed by the anterior wall of the left ventricle turning a pale color. The chest was then closed, and the mice were extubated and allowed to recover from anesthesia. Mice were given post-operative analgesics (buprenorphine-sustained release formula by s.c. injection at 0.1 mg/kg) and allowed to recover until the experimental time points indicated, at which point mice were then further analyzed or tissues were harvested. I/R procedures were performed in the Duke Cardiovascular Physiology Core. LAD ligation procedures were performed in collaboration with Dr. Bursac's lab at Duke University.

## Echocardiography

To assess cardiac function before and after injury, mice were anesthetized by 2% isoflurane inhalation and imaged by B-mode and M-mode echocardiography using a Vevo3100LT instrument with a 25–55 MHz transducer (MX550D, VisualSonics). At least 5x short axis M-mode traces of 8–10 s were recorded across the mid-papillary region of the LV. Systole and diastole LV dimensions were measured using VevoLab's Auto LV analysis tool. All measurements were averaged for each heart at each time point. Heart function was measured by calculating ejection fraction (EF) from the following formula: EF(%)=100*[(LVED–LVES)/LVED], where LVED = [7.0/(2.4+LVIDd)]*(LVIDd)$^3$ and LVES = [7.0/(2.4+LVIDs)]*(LVIDs)$^3$. Abbreviations: LVED, left ventricular end-diastolic volume; LVES, left ventricular end-systolic volume; LVIDd, left ventricular internal dimension at end-diastole; LVIDs, left ventricular internal dimension at end-systole.

## AAV capsid library

AAV9-based capsid libraries were constructed using saturation mutagenesis of residues within the VR-IV site as previously described (*Gonzalez et al., 2023*; *Rosales et al., 2025*). The parental library plasmid, containing AAV2 *ITR*s flanking AAV2 *Rep* and AAV9 *Cap* genes (pTR-wtAAV9-VR-IV), was generated with degenerate primers flanking the VR-IR region (amino acids 452–458 in *Cap*). Prior to saturation mutagenesis with degenerate primers, the VR-IV region of *Cap* was replaced with a partial *GFP* sequence to prevent incorporation of wild-type VR-IV sequences in the parental library. The parental library insert was incorporated into the pTR-wtAAV9-VR-IV plasmid to replace the *GFP* sequence via restriction enzyme digestion with XbaI and BsiWI followed by ligation (New England Biolabs). Purified ligation products were electroporated into DH10B ElectroMax cells (Invitrogen #18290015) and immediately plated on 245 mm Square BioAssay dishes (Corning #431111) containing LB agar with ampicillin. Plates were incubated at 32°C overnight to allow colony growth and formation. The following day colonies were pooled and harvested for plasmid DNA collection via ZymoPURE plasmid MaxiPrep kit (Zymo Research, #D402). Purified library plasmids were co-transfected with adenovirus helper plasmid into adherent HEK293 cells to generate AAV9 capsid libraries as described below.

## AAV vector production and dosing

Recombinant AAV vectors and libraries were prepared using adherent or suspension HEK293 cell cultures as previously described (*Gonzalez et al., 2022*; *Gonzalez et al., 2023*; *Rosales et al., 2025*). For AAV libraries, at 80–90% confluency in 150 cm tissue culture dishes (Fisher Scientific, #087726), cells were triple transfected via polyethylenimine (PEI, Sigma-Aldrich) in a 1:3 DNA:PEI ratio, containing transfer plasmid (pITR), adenovirus helper plasmid (pXX680), and RepCap plasmid (AAV9, AAV.cc84, or AAV.IR41). Cell media was harvested at days 4 and 6 post-transfection. For

recombinant AAV vectors, suspension HEK293 cells were used. HEK293 cells were originally obtained from the UNC Vector Core and authenticated by whole-genome sequencing. Similar to adherent transfection, suspension cells were triple-plasmid-transfected with PEI, containing pITR (0.3 ug/mL), pXX680 (0.6 ug/mL), and RepCap (0.5 ug/mL). Media was harvested at 6 days post-transfection with cells pelted and discarded by centrifugation. Both adherent and suspension AAV packaging work-flows follow the same downstream purification processes. Briefly, harvested media was incubated with 12% polyethylene glycol (PEG) overnight at 4°C. Media was then centrifuged at 4000 × $g$ at 4°C for 45 min to precipitate PEG containing AAV. PEG was resuspended in AAV formulation buffer, containing 1 mM $MgCl_2$ and 0.001% F-68 in dPBS. Resuspended PEG and AAV was DNase treated at 37°C for 1 h, followed by iodixanol gradient purification via ultracentrifugation at 30,000 RPM at 17°C overnight. Iodixanol gradients consisted of 60%, 40%, 25%, and 17% densities. Following centrifugation, 550 ul fractions were collected starting from the 25–40% border and ending at the 40–60% border. All fractions were titered for AAV by qPCR using primers targeting the ITR region (see Key Resources Table). Iodixanol fractions containing the highest titer were selected for downstream desalting and buffer exchange. Desalting was performed using manufacturer's instructions with a PD MidiTrap G-25 column (Cytiva). Following desalting, buffer exchange with sterile AAV formulation buffer was performed according to manufacturer's instructions with a Pierce high-capacity endotoxin removal spin column (Thermo Scientific). Purified final AAV titer was measured by QIAcuity Digital PCR (dPCR) system with QIAcuity probes targeting AAV2-ITR (QIAGEN, #250102). Titered AAV was systemically delivered in 8–12-week-old mice via retro-orbital injection. AAV dose was calculated as viral genomes per kilogram body weight to account for body size differences across age and sex.

## In vivo AAV capsid library screening in mice

The parental AAV9 capsid library for the VR-IV region was titered by qPCR using primers targeting the ITR region (see Key Resources Table). Parental libraries were systemically delivered to adult C57BL/6J mice (The Jackson Laboratory, #000664) that had undergone sham (n=1) or I/R (n=2) surgeries at 9 dpi. Sufficient cardiac injury by I/R was confirmed by echocardiography measurements of left ventricular ejection fraction. All mice were dosed with $2.5 \times 10^{13}$ v.g./kg b.w. of the parental library. Two days after library delivery, at 11 dpi, all mice were sacrificed, and hearts were separately biopsied at the injury (infarct and border zone) and remote sites (right ventricle). Sham mice had heart biopsies captured from approximately the same areas, although no infarct was present. Injured and remote collected tissues were digested with Proteinase K for DNA isolation using a PureLink Genomic DNA mini kit (Invitrogen #K182002). AAV genomes were amplified from the genomic DNA using PCR primers targeting the AAV9 *Cap* gene (see Key Resources Table). Amplified AAV9 *Cap* sequences were then ligated into our AAV library plasmid backbone (pTR-wtAAV9-VR-IV) and used to generate a second AAV library. The parental and second viral libraries for all biopsies (remote, injured) and mice (sham, I/R) were prepared for high-throughput sequencing to assess sequence diversity and enrichment. All viral libraries were Dnase I-treated prior to extraction of viral genome from the capsid, and subsequently fortified with Illumina adapter and index sequences by PCR. All PCR products were purified using the PureLink PCR micro kit (Invitrogen). Prepared libraries were then prepared for sequencing with the Illumina NovaSeq 6000 S-Prime Reagent Kit and sequenced with the Illumina NovaSeq system. Demultiplexed reads were analyzed with an updated in-house Perl script (*Gonzalez et al., 2022*; *Gonzalez et al., 2023*; *Rosales et al., 2025*). Reads are surveyed for nucleotide sequences flanking the AAV9 VR-IV library region and intermediate sequences are counted and ranked. These nucleotide sequences are then translated, and the resulting amino acid sequences are also counted and ranked. Fold change enrichment in the injury versus remote biopsy was calculated using normalized read counts for each amino acid sequence. Co-expressed amino acid sequences found in both I/R-injured mice were plotted against each other to find amino acid sequences with >100-fold injury enrichment in I/R-injured mice. Injury fold change over remote was then averaged for both I/R-injured mice and plotted against the injury fold change for the corresponding amino acid sequence found in the sham mouse. Scatterplots were generated using the RStudio graphics package v3.5.2.

## IVIS imaging

All IVIS bioluminescence imaging was conducted with adult albino BALB/c mice (Charles River) using equipment provided by the Duke Optical Molecular Imaging and Analysis shared resource (OMIA)

core. D-luciferin, potassium salt (Gold Biotechnology), was dissolved in dPBS (without calcium or magnesium) at a final concentration of 15 mg/mL. The D-luciferin was sterile-filtered with a 0.22 um filter, and aliquots were stored frozen for future use. At each imaging timepoint, D-luciferin was thawed on ice in a covered container to protect from light. While mice were anesthetized under 2% isoflurane, thawed D-luciferin was injected intraperitoneally at 150 mg/kg b.w. dose. Regions of interest for this study, such as the heart and liver, had body hair above them removed using Nair to improve signal-to-noise ratio. Luciferase signals were allowed a 10–15 min plateau time. Bioluminescence images were then collected using an IVIS Kinetic instrument and Living Image Software (PerkinElmer). Average and maximum radiances were quantified in specified ROIs with the Living Image software. For ex vivo bioluminescence imaging, mice were sacrificed and organs immediately harvested for imaging. Organs were quickly washed with ice-cold PBS and DMEM to remove blood and debris. After washing, organs were submerged in room temperature D-luciferin solution for 10 min, then immediately placed on black construction paper and imaged in the IVIS Kinetic.

## Viral genome biodistribution

Genomic DNA was extracted from tissues using a PureLink Genomic DNA mini kit (Invitrogen #K182002) following manufacturer's instructions. Viral genomes were then quantified by qPCR with a known DNA mass of pITR transfer plasmid standard used for AAV packaging. qPCR primers were designed to target the border region between the transgene and bgh-PolyA sequence (below). Quantified viral genomes are presented as a ratio of viral genomes per microgram of extracted genomic DNA used for the reaction.

Primers used in this study are listed in the Key Resources Table.

## Immunofluorescence

Adult mouse hearts were immediately harvested after sacrifice, perfused with room temperature dPBS (without calcium or magnesium), and fixed overnight in 4% paraformaldehyde (Fisher #50-980-487) at 4°C. After fixing, hearts were washed in PBS three times for 15 min each and incubated in 30% sucrose in PBS overnight at 4°C. Hearts were then transferred and embedded in Tissue-Plus OCT (Fisher #23-730-571). Tissue blocks were cryosectioned at 10 um and placed on a Superfrost Plus Microscope slide (Fisher #22-037-246). Slides were stored at –80°C until staining. On the day of staining, slides were thawed on a slide warmer and allowed to dry for 30 min. Tissues were rehydrated with PBS, washed 3× with PBS-T (PBS +0.1% Triton X100), and then blocked for 1 h at room temperature in PBS-T with 5% normal goat serum. After blocking, slides were incubated with primary antibodies diluted in antibody solution (PBS-T + 1% BSA) overnight in a humidified chamber at 4°C. Primary antibodies included: anti-GFP (Invitrogen #A-11122) at 1:1000 dilution and anti-vimentin (DSHB #40E-C) at 1:100 dilution. Following primary antibody incubation, slides were washed 3× with PBS-T and incubated with secondary antibodies, DAPI (Thermo Scientific #62248, 1:1000 dilution), and Alexa Fluor 546 Phalloidin (Invitrogen #A22283, 1:200 dilution) in antibody solution for 2 h at room temperature. Secondary antibodies included goat anti-rabbit IgG Alexa Fluor 488 (Invitrogen #A-11008) at 1:500 dilution and goat anti-mouse IgG Alexa Fluor 555 (Invitrogen #A-21422) at 1:500 dilution. Following incubation, slides were washed 3× with PBS for 5 min each and mounted with Vectashield HardSet Antifade Mounting media (Vector Laboratories #H-1400–10). Mounted slides were stored at 4°C until imaged.

## Acknowledgements

We thank the Duke Cardiovascular Physiology Core for cardiac injuries. We acknowledge NIH for research funding (NB: R01EB032726, R01HL164013, and U01HL134764; AA: R01HL089221, R01DK134408; KDP: R35HL150713).

# Additional information

## Funding

| Funder | Grant reference number | Author |
|---|---|---|
| National Heart, Lung, and Blood Institute | R35HL150713 | Kenneth D Poss |
| National Heart, Lung, and Blood Institute | R01HL164013 | Nenad Bursac |
| National Heart, Lung, and Blood Institute | U01HL134764 | Nenad Bursac |
| National Heart, Lung, and Blood Institute | R01HL089221 | Aravind Asokan |
| National Institute of Biomedical Imaging and Bioengineering | R01EB032726 | Nenad Bursac |
| National Institute of Diabetes and Digestive and Kidney Diseases | R01DK134408 | Aravind Asokan |

The funders had no role in study design, data collection and interpretation, or the decision to submit the work for publication.

## Author contributions

David W Wolfson, Conceptualization, Formal analysis, Investigation, Visualization, Methodology, Writing – original draft, Writing – review and editing; Joshua A Hull, Software, Formal analysis, Methodology; Yongwu Li, Valentina Cigliola, Kelsey A Oonk, Methodology; Trevor J Gonzalez, Alan Rosales, Formal analysis, Methodology; Mourya D Jayaram, Formal analysis, Vector production; Garth W Devlin, Formal analysis, Vector production; Nenad Bursac, Conceptualization, Resources; Aravind Asokan, Conceptualization, Resources, Formal analysis, Supervision, Funding acquisition, Writing – review and editing; Kenneth D Poss, Conceptualization, Resources, Supervision, Funding acquisition, Project administration, Writing – review and editing

## Author ORCIDs

David W Wolfson ⓘ https://orcid.org/0000-0003-3850-3925
Nenad Bursac ⓘ https://orcid.org/0000-0002-5688-6061
Kenneth D Poss ⓘ https://orcid.org/0000-0002-6743-5709

## Ethics

All mouse protocols were approved by the Institutional Animal Care and Use Committee (IACUC) at Duke University (Protocols #A003-22-01, #A025-24-01), which is accredited by the Association for Assessment and Accreditation of Laboratory Animal Care-International (AAALAC).

Reviewer #1 (Public review): https://doi.org/10.7554/eLife.107148.3.sa1
Reviewer #2 (Public review): https://doi.org/10.7554/eLife.107148.3.sa2
Reviewer #3 (Public review): https://doi.org/10.7554/eLife.107148.3.sa3
Author response https://doi.org/10.7554/eLife.107148.3.sa4

# Additional files

## Supplementary files

MDAR checklist

## Data availability

Data are deposited to Dryad, and materials generated in this manuscript are maintained by the corresponding authors and available by request.

The following dataset was generated:

| Author(s) | Year | Dataset title | Dataset URL | Database and Identifier |
|---|---|---|---|---|
| Wolfson D, Hull J, Li Y, Gonzalez T, Jayaram M, Devlin G | 2026 | Data from: Spatial and longitudinal tracking of enhancer-AAV vectors that target transgene expression to injured mouse myocardium | https://doi.org/10.5061/dryad.2547d7x4p | Dryad Digital Repository, 10.5061/dryad.2547d7x4p |

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
