## [Editor Report · eLife Assessment]

This study identifies novel approaches to improving transgene expression in the injured mammalian myocardium through a combination of a tissue regeneration enhancer element and engineered AAVs - specifically, a liver-detargeting capsid, AAV.cc84, and an in vivo library screen-selected AAV-IR41. The evidence is **convincing**, and the AAV vectors are of **fundamental** value to the field of cardiac gene therapy. Future research exploring how to combine the features of AAV.cc84 and AAV-IR41 could yield an even more promising vector for therapeutic use.

---

## [Referee Report · Reviewer #1 (Public review)]

In this manuscript, Wolfson and co-authors demonstrate a combination of an injury-specific enhancer and engineered AAV that enhances transgene expression in injured myocardium. The authors characterize spatiotemporal dynamics of TREE-directed AAV expression in the injured heart using a non-invasive longitudinal monitoring system. They show that transgene expression is drastically increased 3 days post-injury, driven by 2ankrd1a. They reported a liver-detargeted capsid, AAV cc.84, with decreased viral entry into the liver while maintaining TREE transgene specificity. They further identified the IR41 serotype with enhanced transgene expression in injured myocardium from AAV library screening. This is an interesting study that optimizes the potential application of TREE delivery for cardiac repair.

Comments on revisions:

The authors are responsive and have addressed my concerns.

---

## [Referee Report · Reviewer #2 (Public review)]

Summary:

In this manuscript by Wolfson et al., various adeno-associated viruses (AAVs) were delivered to mice to assess the cardiac-specificity, injury border-zone cardiomyocyte transduction rate, and temporal dynamics in the goal to find better AAVs for gene therapies targeting the heart. The authors delivered tissue regeneration enhancer elements (TREEs) controlling luciferase expression and used IVIS imaging to examine transduction in the heart and other organs. They found that luciferase expression increased in the first week after injury when using AAV9-TREE-Hsp68 promoter, waning to baseline levels by 7 weeks. However, AAV9 vectors transduced the liver, which was significantly reduced by using an AAV.cc84 liver de-targeting capsid. The authors then performed in vivo screening of AAV9 capsids and found AAV-IR41 to preferentially transduce injured myocardium when compared to AAV9. Finally, the authors combined TREEs with AAV-IR41 to show improved luciferase expression compared to AAV9-TREE at 7, 14 and 21 days after injury.

Overall, this manuscript provides insights into TREE expression dynamics when paired with various heart-targeting capsids, which can be useful for researchers studying ischemic injury of murine hearts. While the authors have shown the success of using AAV9-TREEs in porcine hearts, it is unknown whether the expression dynamics would be similar in pigs or humans, as mentioned in the limitations.

Strengths:

Important contribution to the AAV gene therapy literature.

Comments on revised version:

My concerns have been adequately addressed.

---

## [Referee Report · Reviewer #3 (Public review)]

Summary:

The tissue regeneration enhancer elements (TREEs) identified in zebrafish have been shown to drive injury-activated temporal-spatial gene expression in mice and large animals. These findings increase the translational potential of findings in zebrafish to mammals. In this manuscript, the authors tested TREEs in combination with different adeno-associated viral (AAV) vectors using in vivo luciferase bioluminescent imaging that allows for longitudinal tracking. The TREE-driven luciferase delivered by a liver de-targeted AAV.cc84 decreased off-target transduction in liver. They further screened an AAV library to identify capsid variants that display enhanced transduction for infarcted myocardium post ischemia reperfusion and myocardial infarction. A new capsid variant, AAV.IR41, was found to show increased transduction post I/R and MI.

Strengths:

The authors injected AAV-cargo several days after ischemia/reperfusion (I/R) injury as a clinically relevant approach. Overall, this study is significant in that it identifies new AAV vectors that can be used to deliver promising genes as potential new gene therapies in the future. The manuscript is well-written and the data are also of high quality.

Weaknesses:

The authors have addressed my previous concerns.

---

## [Author Response]

The following is the authors’ response to the original reviews.

**Reviewer #1 (Public review):**
In this manuscript, Wolfson and co-authors demonstrate a combination of an injury-specific enhancer and engineered AAV that enhances transgene expression in injured myocardium. The authors characterize spatiotemporal dynamics of TREE-directed AAV expression in the injured heart using a non-invasive longitudinal monitoring system. They show that transgene expression is drastically increased 3 days post-injury, driven by 2ankrd1a. They reported a liver-detargeted capsid, AAV cc.84, with decreased viral entry into the liver while maintaining TREE transgene specificity. They further identified the IR41 serotype with enhanced transgene expression in injured myocardium from AAV library screening. This is an interesting study that optimizes the potential application of TREE delivery for cardiac repair. However, several concerns were raised prior to publication:Major Concerns:(1) In Figure 1, the authors demonstrated that 2andkrd1aEN is not responsive to sham injury after AAV delivery, but Figure 3 shows a strong response to sham when AAV is delivered after injury. The authors do not provide an explanation for this observation.

This discrepancy is due to the timing of AAV delivery. In Figure 1, AAV was delivered 60 days prior to IVIS imaging and cardiac injury, allowing time for the baseline level of AAV transgene expression to reach a plateau. From this baseline level, we were able to measure fold change in luminescence signal before and after cardiac injury. In Figure 3, AAV was delivered 4 days after cardiac injury. Luminescence in the heart was measured 3 days later (day 7), when the baseline of AAV transgene expression is still building. The data from Figure 1C-D inform us that the 2ankrd1aEN response to cardiac injury peaks within the first week and returns to baseline levels after 5-7 weeks. In Figure 3E, we show that 2ankrd2aEN provides a baseline level of expression that is present in sham hearts and reaches its plateau after 6 weeks. In contrast, I/R injured hearts show enhanced expression in the first 3-4 weeks, corresponding with the dynamics of 2ankrd1aEN’s response to injury observed in Figure 1C. We have now included a phrase in the revised manuscript on p. 7, paragraph 1 to clarify.

(2) In Figure 4, a higher GFP signal is observed in all areas of the heart of the IR41-treated mouse compared to AAV9. The authors should compare GFP expression between AAV9 and IR41 in uninjured hearts and provide insights into enhanced cardiac tropism to confirm that IR41 is MI injury enriched, not Sham as well.

We sought to address this question with the experiments presented in Figure 5. We treated sham mice with AAV9 and IR41 containing 2ankrd1aEN. Figure 5D showed IR41 delivered more vector genomes to the sham heart on average, though not with a p-value less than 0.05 compared with AAV9. In Supplemental Figure 5B, IR41 also provided higher luminescence at day 7 post-sham but was comparable at day 14 and day 21. These data suggest IR41 might increase heart tropism in healthy hearts, but IR41’s effect is most dramatic when delivered to injured hearts, where cardiac vector genomes are highest (Figure 5D). We have now included a sentence in the revised manuscript on p. 8, paragraph 2 to clarify.

(3) The authors should clarify which model is being used between myocardial infarction (MI) and Ischemia-reperfusion (IR) throughout the figures, as the experimental schemes and figure legends did not match with each other (MI or IR in Figure 1A, 1D, 3A, and 3E). Both models cause different types of injuries. The authors should explain the difference in TREE expression in both models.

We have revised the figures to specify the model, where I/R or MI is used.

(4) In Figure 2, the authors use REN instead of 2ankrd1aEN to demonstrate liver-detargeting using AAV cc.84. Is there a specific reason?

Our data in Figure 1 informed us that off-target liver expression is more specifically an issue for REN compared to 2ankrd1aEN. Baseline levels of luminescence in the heart could not be as clearly marked due to off-target expression in the liver, which was showcased in Figure 2B with AAV9 delivery to sham mice. As discussed above, 2ankrd1aEN provided stronger baseline levels of expression of the heart which could be more clearly marked in IVIS images for tracking fold changes over time. For these reasons, we sought to explore how incorporation of the AAV.cc84 capsid could be utilized to minimize off-target liver expression. We have now included a sentence in the revised manuscript on p. 5, paragraph 3 to clarify.

**Reviewer #2 (Public review):**
In this manuscript by Wolfson et al., various adeno-associated viruses (AAVs) were delivered to mice to assess the cardiac-specificity, injury border-zone cardiomyocyte transduction rate, and temporal dynamics, with the goal of finding better AAVs for gene therapies targeting the heart. The authors delivered tissue regeneration enhancer elements (TREEs) controlling luciferase expression and used IVIS imaging to examine transduction in the heart and other organs. They found that luciferase expression increased in the first week after injury when using AAV9-TREE-Hsp68 promoter, waning to baseline levels by 7 weeks. However, AAV9 vectors transduced the liver, which was significantly reduced by using an AAV.cc84 liver de-targeting capsid. The authors then performed in vivo screening of AAV9 capsids and found AAV-IR41 to preferentially transduce injured myocardium when compared to AAV9. Finally, the authors combined TREEs with AAV-IR41 to show improved luciferase expression compared to AAV9-TREE at 7, 14, and 21 days after injury.Overall, this manuscript provides insights into TREE expression dynamics when paired with various heart-targeting capsids, which can be useful for researchers studying ischemic injury of murine hearts. While the authors have shown the success of using AAV9-TREEs in porcine hearts, it is unknown whether the expression dynamics would be similar in pigs or humans, as mentioned in the limitations.The following questions and concerns can be addressed to improve the manuscript:(1) From the IVIS data, it seems that the Hsp68 promoter might not be "normally silent in mouse tissues," specifically in the liver (Figure S1B). Are there any other promoters that can be combined with TREEs to induce cardiac-injury specific expression while minimizing liver expression? This could simplify capsid design to focus on delivery to injured areas.

Indeed we found the Hsp68 promoter does provide low levels of baseline expression, especially in the liver of mice. The Hsp68 promoter was initially chosen due to its permissive nature allowing for assessment of expression directed by TREEs. Many or most groups use the Hsp68 promoter for enhancer tests in mice, but we agree that other permissive promoters might have lower baseline levels of expression and might have the benefit of smaller size. We have not rigorously tested other permissive promoters in our experiments.

(2) Why is it that AAV9-TREE-Hsp68-Luc wane in expression (Figure 1C and 1D), whereas AAV.cc84-TREE-Hsp68-Luc expresses stably for over 2 months (3E)? This has important implications for the goal of transience in gene delivery.

Please see our response to reviewer 1’s comment #1 above.

(3) AAV-IR41 was found to transduce cardiomyocytes in the injured zone. However, this capsid also shows a very strong off-target liver expression. From a capsid design perspective, is it possible to combine AAV-cc84 and AAV-IR41?

This approach is in theory possible as these epitopes are structurally distinct. However, since the mechanism (receptor usage) is currently unknown, it would not be possible to predict whether the properties are mutually exclusive. Further, we would need to ensure that combining modifications does not impact vector yield. We can explore such features with next generation candidates as we continue to improve the platform. We have now included a sentence in the revised manuscript on p. 9, paragraph 3, mentioning the possibility of combining the two capsid mutations.

(4) It would be helpful to see immunostaining for the various time points in Figure 5. Is it possible to use an anti-luciferase antibody (or AAV-TREE-Hsp68-eGFP) to compare the two TREE capsids?

We were not able to do immunostaining of luciferase expression, because the biopsied hearts were used to quantify vector genomes via qPCR. We have previously reported results of immunostaining of EGFP expression directed by 2ankrd1aEN in I/R-injured mouse hearts (Yan et al., 2023), which we expect to match the expression seen in these experiments.

**Reviewer #3 (Public review):**
Summary:The tissue regeneration enhancer elements (TREEs) identified in zebrafish have been shown to drive injury-activated temporal-spatial gene expression in mice and large animals. These findings increase the translational potential of findings in zebrafish to mammals. In this manuscript, the authors tested TREEs in combination with different adeno-associated viral (AAV) vectors using in vivo luciferase bioluminescent imaging that allows for longitudinal tracking. The TREE-driven luciferase delivered by a liver de-targeted AAV.cc84 decreased off-target transduction in the liver. They further screened an AAV library to identify capsid variants that display enhanced transduction for myocardium post-myocardial infarction. A new capsid variant, AAV.IR41, was found to show increased transduction at the infarct border zones.Strengths:The authors injected AAV-cargo several days after ischemia/reperfusion (I/R) injury as a clinically relevant approach. Overall, this study is significant in that it identifies new AAV vectors for potential new gene therapies in the future. The manuscript is well-written, and their data are also of high quality.Weaknesses:The authors might be using MI (myocardial infarction) and I/R injury interchangeably in their text and labels. For instance, "We systemically transduced mice at 4 days after permanent left coronary artery ligation with either AAV9 or IR41 harboring a 2ankrd1aEN-Hsp68::fLuc transgene. IVIS imaging revealed higher expression levels in animals transduced with IR41 compared to AAV9, in both sham and I/R groups (Fig. 5A)". They should keep it consistent. There is also no description for the MI model.

We have adjusted figure labels and main text to ensure the injury model is described correctly.

We have also addressed all additional Recommendations for the authors, which requested minor modifications to figures like error bars and image annotation.